# Determination of Hexabromocyclododecane in Expanded Polystyrene and Extruded Polystyrene Foam by Gas Chromatography-Mass Spectrometry

**DOI:** 10.3390/molecules26237143

**Published:** 2021-11-25

**Authors:** Tianao Mao, Haoyang Wang, Zheng Peng, Taotao Ni, Tianqi Jia, Rongrong Lei, Wenbin Liu

**Affiliations:** 1Hangzhou Institute for Advanced Study, University of Chinese Academy of Sciences, Hangzhou 310024, China; maotianao@foxmail.com; 2Research Center for Eco-Environmental Sciences, Chinese Academy of Sciences, Beijing 100085, China; ttaoni@yeah.net (T.N.); tqijia@126.com (T.J.); leirongr@163.com (R.L.); 3Environmental Protection and Foreign Cooperation and Exchange Center of Ministry of Ecology and Environment, Beijing 100035, China; peng.zheng@fecomee.org.cn; 4University of Chinese Academy of Sciences, Beijing 100049, China

**Keywords:** gas chromatography-mass spectrometry, HBCD, EPS, XPS, POPs

## Abstract

A gas chromatography-mass spectrometry (GC/MS) method for the determination of hexabromocyclododecane (HBCD) in expanded polystyrene and extruded polystyrene foam (EPS/XPS) was developed. The EPS/XPS samples were ultrasonically extracted with acetone and the extracts were purified by filtration through a microporous membrane (0.22 μm) and solid-phase extraction. The samples were analyzed using a GC/MS using the selected ion monitoring mode. The ions 157, 319 and 401 were selected as the qualitative ions, while ion 239 was chosen as the quantitative ion. An HBCD standard working solution with a concentration range of 1.0–50.0 mg/L showed good linearity. The detection limit of HBCD was 0.5 mg/kg, meeting the LPC limit (<100 or 1000 mg/kg). Six laboratories were selected to verify the accuracy of the method, and 10 samples were tested. The interlaboratory relative standard deviation range was 3.68–9.80%. This method could play an important role in controlling HBCD contamination in EPS/XPS.

## 1. Introduction

Hexabromocyclododecane (HBCD) is flame retardant with high bromine content that has long been used in the manufacture of expanded polystyrene (EPS) and extruded polystyrene (XPS) boards for fire protection and insulation in buildings. In the 1980s, HBCD was detected in air, sludge and sediment in Sweden [1]. Since then, many researchers in other countries have also confirmed the widespread presence of HBCD in soils [2], outdoor air [3,4], seawater [5], and house dust [6]. Also, HBCD has been detected in arctic regions [7] and breast milk [8]. There is global concern about the potential toxic effects of HBCD on humans and ecosystems [3,9] because of its persistence in the environment, bioaccumulation, and bioamplification in fish [10], birds [11], and mammals [12,13]. Recently many studies in animals have shown that HBCD can affect the expression of related genes in rats [14] and zebrafish [10]. Furthermore, HBCD promoted the production and accumulation of fat in vivo and in vitro [15]. HBCD was added to the control list in the Stockholm Convention on Persistent Organic Pollutants in May 2013 and banned from future production and use [16].

The latest information on global production of HBCD indicates that total production in 2011 was estimated at 31,000 metric tons, and it was mainly produced in China, Europe, and the United States. Considering that 90% of HBCD on average is used in the manufacture of EPS/XPS each year [17], large quantities of HBCD-containing EPS/XPS building materials have accumulated globally over the past decade. Because of the long service life of EPS/XPS (20–50 years) [18,19], many of these building materials are still in use. However, a study shows that the use of HBCD in some consumer products is unregulated [20], and there is still a risk of HBCD entering the environment from existing construction materials and waste through wear and tear during product use, weathering, and leachate from landfill [21]. As of 2019, HBCD must be destroyed or irreversibly transformed to prevent its entry into the environment when it is at or above the designated LPC limits (100 or 1000 mg/kg) in accordance with the European Union’s Basel Convention General Technical Guidelines for Persistent Organic Pollutants Waste Management [22]. Therefore, a simple and quick method is required to determine whether HBCD in EPS/XPS exceeds the standard (LPC). Many methods have been developed to determine HBCD concentration in EPS or XPS foam like X-ray fluorescence spectroscopy (XRF), flowing atmospheric pressure afterglow mass spectrometry, and liquid chromatography-tandem mass spectrometry (LC-MS/MS) [23,24,25,26,27,28].

In this study, the GC/MS method was developed to determine the HBCD concentration in EPS/XPS products. The solvent and purification processes were investigated to separate the HBCD and EPS/XPS matrix in the extract dilution. Six laboratories were selected to carry out the verification work, and 10 samples were tested to verify the feasibility of the method. This study aimed to establish a simple, inexpensive and effective analytical method for determination of HBCD in EPS/XPS. It is used to help more countries and regions judge whether HBCD in construction waste exceeds LPC limits, and provides a reference for the recycling or destruction and irreversible transformation of waste containing HBCD.

## 2. Materials and Methods

### 2.1. Samples and Chemicals

HBCD standard (purity >97%) was purchased from Dr. Ehrenstorfer GmbH (Augsburg, Germany). Acetone (pesticide residue grade) was obtained from Fisher Chemical (Thermo Fisher Scientific Inc., Waltham, MA, USA). Toluene and n-Hexane (pesticide residue grade) were purchased from J.T. Baker Corporation (Phillipsburg, NJ, USA). Methanol and dichloromethane (analytical pure) were purchased from Sinopharm (Beijing, China). Polytetrafluoroethylene filtration membrane (0.22 μm) was purchased from Jintengyi Technology Co., Ltd. (Beijing, China). Five EPS and five XPS samples were collected from 10 companies in Shandong China. 20 mg HBCD was dissolved with 2 mL acetone and diluted to 100 mL with n-hexane to prepare the HBCD stock solution (200 mg/L). Working solutions with HBCD concentrations of 1.0, 2.0, 5.0, 10.0, 20.0, and 50.0 mg/L were prepared by dilute with n-hexane of the HBCD stock solution.

### 2.2. Sample Preparation

The EPS/XPS samples were cut into particles smaller than 5 mm. 0.1 g (±0.1 mg) of EPS/XPS sample was weighed from the particles above, then placed in a 10 mL stoppered colorimetric tube, and 5 mL of acetone was added to dissolve the sample. The colorimetric tube was placed on a vortex mixer for 2 min and then extracted by ultrasonication (KQ-100E, Kunshan Ultrasonic Instrument Co., Ltd., Kunshan, China) for 20 min at room temperature. The supernatant was filtered through a 0.22 μm polytetrafluoroethylene filter membrane. The filtrate was collected and n-hexane was added to give a volume of 20 mL. Because the concentration of HBCD in the EPS/XPS samples were usually very high, only part of the filtrate was analyzed. For the EPS sample, the HBCD concentration in typical EPS product was approximately 0.6–0.8%, 2.5% (500 μL) of the filtrate was analyzed. For the XPS sample, the typical HBCD in XPS was approximately 4–6%, 0.25% (50 μL) of the filtrate was analyzed and n-hexane (450 μL) was added to make the volume up to 500 μL. Next, the sample was purified using a solid-phase extraction (SPE) column (LC-Si, 500 mg, 6 mL, Supelco Inc., St. Louis, MO, USA). The SPE column was activated with 6 mL of n-hexane. The EPS/XPS filtrates were eluted once with 6 mL of acetone. All of the eluent was collected and then concentrated under a stream of nitrogen gas. The concentrated samples were reconstituted with n-hexane to 1 mL. Finally, the samples were placed in 2 mL brown vials for GC/MS analysis. An EPS sample without HBCD was selected as the blank sample, extracted, and purified together with other EPS/XPS samples.

### 2.3. GC/MS Analyses

The samples were analyzed using an Agilent 6890A/5973C GC/MS with a DB-5 MS capillary column (Agilent Technologies, Santa Clara, CA, USA). The length of DB-5 MS capillary column is 30 m, the inner diameter is 0.25 mm and the film thickness is 0.25 m. The following heating procedure of oven temperature was applied: initial temperature is 60 °C, held at 60 °C for 2 min, increased to 270 °C at 15 °C/min and held at this temperature for 5 min, then increased to 290 °C at 5 °C/min and held at this temperature for 5 min. The carrier gas used helium (purity ≥99.999%), and the flow rate was 1.0 mL/min. The sampling method was split-less injection and the injection volume was 1 μL. The inlet temperature was 230 °C. The ionization mode was electron ionization, the ionization energy was 70 eV, and the ion source temperature was 230 °C. The quadrupole temperature was 150 °C and the interface temperature was 280 °C. The data acquisition mode used selected ion monitoring. The selected monitoring ions (m/z) are shown in Table 1. The ions 157, 319 and 401 were selected as the qualitative ions, while ion 239 was chosen as the quantitative ion.

### 2.4. Quality Assurance and Quality Control

Three samples of EPS and XPS each were selected to estimate the matrix effect (*ME*) according to the method reported by Caban et al. [29]. The *ME* is calculated according to the following formula.
(1)$$ME=\left(\frac{B-C}{A}-1\right)\times 100\%$$
where *A* is the peak area of the HBCD standard solution (20 mg/L). *B* is the peak area of the EPS/XPS sample with HBCD standard (20 mg/L) added before injection. *C* is the peak area of the non-spiked EPS/XPS sample. The samples showed acceptable matrix effects ranging from −10.9% to 22.0%.

The HBCD was not detected in the blank sample. The recovery of the blank spiked sample test is between 87% and 113%.

## 3. Results and Discussion

### 3.1. Optimization of the Sample Pretreatment Conditions

Because dissolution of polystyrene from EPS/XPS would cause matrix interference in the HBCD analysis, different extraction solvents were screened to optimize the method. Through literature research [23,26,30,31], we chose methanol, toluene, acetone, dichloromethane, n-hexane, n-hexane/isopropanol (1:1, *v*/*v*), acetone/methylene chloride (1:1, *v*/*v*), toluene/methylene chloride (1:1, *v*/*v*), acetone/n-hexane (1:1, *v*/*v*) and n-hexane/methylene chloride (1:1, *v*/*v*) to conduct experiments on the dissolution of EPS/XPS and the extraction effect of HBCD.

EPS/XPS partially dissolved in acetone, n-hexane, and n-hexane/isopropanol, but did not dissolve in methanol. Toluene and dichloromethane completely dissolved EPS/XPS. Although acetone emulsified EPS/XPS, minimal dissolution of EPS/XPS occurred, and this solvent gave a very high HBCD extraction efficiency. Therefore, acetone was selected as the extraction solvent. The extract was then purified using SPE. We found that HBCD was adsorbed by the LC-Si filler in the SPE column, and was completely eluted by acetone. The results of the matrix effect also proved that effectiveness of acetone.

### 3.2. Optimization of the GC/MS Conditions

HBCD can be analyzed using a 15 or 30 m capillary column. We found that a 30 m capillary column gave effective separation of HBCD in EPS/XPS and good peak shapes. The heating procedure was optimized, and the retention time of HBCD was 25.40 min. The injection temperatures are the important variable in the determination of HBCD. When the injection temperature is too low, it will lead to incomplete gasification of HBCD and reduce the amount available for detection. If the injection temperature is too high, it will cause thermal isomerisation and degradation of HBCD [32,33,34]. In a comparison of the chromatograms obtained with injection temperatures of 190 °C, 230 °C, and 270 °C, the best results were obtained at 230 °C (Figure 1). At 190 °C, the gasification of HBCD was incomplete and the response was low, which was only 20–25% of those at 230 °C. At 270 °C, the HBCD was decomposed, the impurity peak (retention time from 17.40 min to 18.6 min) increased five times of those at 230 °C. Consequently, 230 °C was selected as the optimum injection temperature, and a good response was obtained using optimized GC/MS conditions.

### 3.3. Linear Range and Detection Limit

The HBCD working solutions were analyzed using the optimized GC/MS conditions. A 1.0 μL aliquot of each HBCD standard working solution was injected. The retention times and chromatographic peak areas were recorded and a standard curve was plotted (Figure 2).

The linear equation for the standard curve of HBCD was *Y* = 1197.6*X* − 283.1, and the *R*^2^ was 0.9934. These results met the test requirements for HBCD analysis [35].

HBCD in real samples were accurately quantified by using a calibration curve. According to the HBCD working curve, the concentration of HBCD in the sample is calculated according to the following formula.
(2)$$\rho =\frac{\;C\times V_{1}\times V_{3}\;\;}{V_{2}\times M}\times D$$
where *ρ* is the concentration of HBCD in EPS or XPS samples (mg/kg). *C* is concentration of HBCD in the sample solution (mg/L). *V*_1_ is volume of extract (ml), *V*_2_ is volume of extracted solution transferred (ml) and *V*_3_ is volume of injection solution (ml). *M* is sample mass (g). *D* is dilution multiple.

The detection limit was determined as described in the references [35]. Seven blank spiked samples with HBCD concentration of 5 mg/kg were continuously tested. The method detection limit is then calculated as 3.143 times the standard deviation. In the present study, the limit of quantitation of detection is 0.5 mg/kg.

### 3.4. Method Precision and Accuracy

To evaluate the method precision and accuracy [35], three blank samples were spiked with HBCD standard solutions to give concentrations of 5, 10, and 20 mg/kg. The HBCD concentration in each of these samples was determined six times. The 18 analysis results were shown in Table 2. The relative standard deviation (RSD) range for the three blank samples was 6.64–6.92%, and the recovery range was 87–113%.

We compared the results from our study with those from previous studies (Table 3). The accuracy of our method met the test requirements.

### 3.5. Actual Sample Analysis and Interlaboratory Verification

To verify the effectiveness of the method, five EPS and five XPS samples were collected from 10 companies. Five verification laboratories (L1–L5) and our laboratory (L6) were commissioned to carry out verification tests for the developed method. The HBCD analysis results from the six laboratories were shown in Figure 3.

For the five EPS samples, the HBCD concentration range was 4.5–6.7 g/kg and the RSD range was 3.78–9.76%. For the five XPS samples, the HBCD concentration range was 29.8–35.5 g/kg, and the RSD range was 3.68–9.80%. The overall RSD range of the 10 EPS/XPS samples was 3.68–9.80% and the R^2^ of the calibration curve of six laboratories were between 0.991 and 0.998.

Through our study, we found that the content of HBCD in EPS/XPS is approximately 1%, with a range of 0.45–0.67% in EPS and 2.98–3.55% in XPS. Such a high concentration of HBCD may be dangerous to human health [36] and pose a challenge to the treatment of construction waste containing EPS/XPS [27,28]. Improper handling methods may cause HBCD to leak into the environment, causing serious damage to the ecological environment [37]. Production of HBCD-containing EPS/XPS was expected to increase by 2.7 million tons per year until 2013, and then decrease by about 7% (18,000 tons) after the inclusion of HBCD in the Stockholm Convention. Because HBCD-containing EPS/XPS products have been produced for more than 40 years, we can infer that 40 to 70 million tons of these products have been produced globally [38]. Consequently, a method is needed to determine HBCD levels in these plastics to provide a basis for the recycling and disposal of HBCD.

## 4. Conclusions

A method was established and verified for the determination of HBCD in the product and waste of EPS/XPS by GC/MS. Satisfactory recovery and precision results were obtained for 10 EPS/XPS samples. The advantages of this method are its high precision, low detection limit, simplicity, speed, and suitability for the detection of HBCD in EPS/XPS samples. Although HBCD has recently been banned, EPS/XPS products containing HBCD are still used on the surface of buildings, and will last for 20–50 years until the demolition of these structures. In the future, HBCD in EPS/XPS waste generated from the demolition of these buildings needs to be detected and analyzed. Therefore, the detection of HBCD in EPS/XPS samples by GC/MS still has great application value.

## Figures and Tables

**Figure 1 molecules-26-07143-f001:**
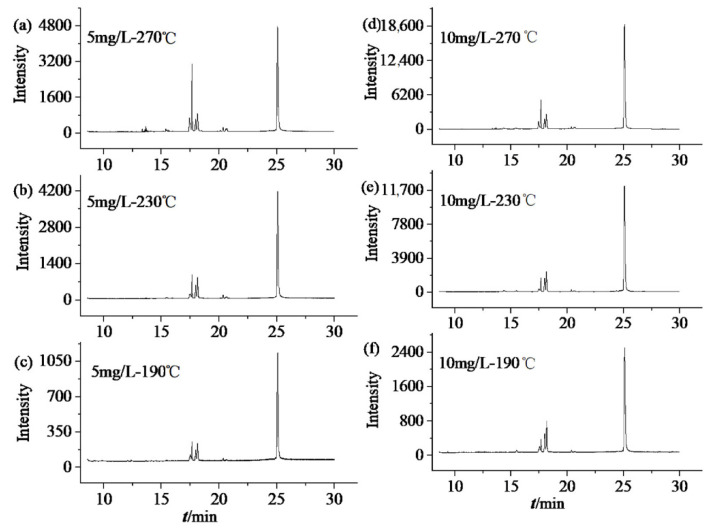
Chromatograms of HBCD at different injection temperatures, (**a**) 5 mg/L-270 °C, (**b**) 5 mg/L-230 °C, (**c**) 5 mg/L-190 °C, (**d**) 10 mg/L-270 °C, (**e**) 10 mg/L-230 °C, and (**f**) 10 mg/L-190 °C.

**Figure 2 molecules-26-07143-f002:**
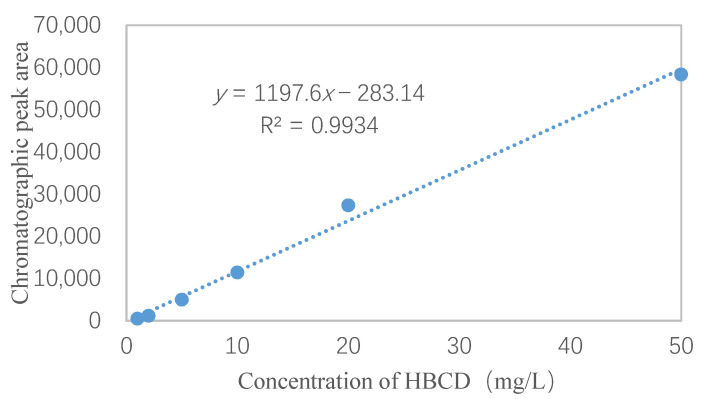
The calibration curve of HBCD.

**Figure 3 molecules-26-07143-f003:**
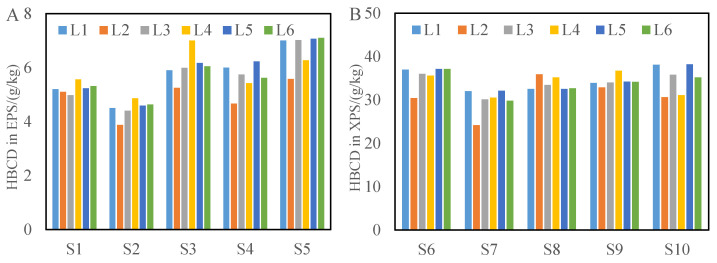
Results from the six laboratories for EPS (**A**) and XPS (**B**).

**Table 1 molecules-26-07143-t001:** Selected monitoring ions (m/z) and allowable relative deviations.

Monitoring of the Ion (m/z)	Ion Species	Ion Ratio (%)	Permissible Relative Deviation (%)
157	Qualitative ion	100	-
239	Quantitative ion	89	±15
319	Qualitative ion	55	±20
401	Qualitative ion	19	±20

**Table 2 molecules-26-07143-t002:** Test data for the method precision and accuracy (n = 6).

HBCD Add the Amount	5 mg/kg	10 mg/kg	20 mg/kg
Measured mean ${\bar{x}}_{i}$	4.9	9.9	21.3
The standard deviation S_i_	0.34	0.69	1.4
Relative standard deviation RSD (%)	6.90%	6.92%	6.64%
Recovery range	87~106%	89~108%	96~113%

**Table 3 molecules-26-07143-t003:** Comparison of experimental results from different studies.

Test Material	Testing Equipment	Concentrations g/kg	RSDs %	Limit of Quantitation mg/kg	The Literature
EPS/XPS	GC/MS	4.5~33.5	6.64~6.92	0.5	In this study
EPS/XPS	XRF	5~12	7~16	50	[23]
EPS	LC-MS/MS	6.849~7.105	0.2~0.6	0.005	[26]
EPS/XPS	XRF, LC-MS/MS, NMR	6.7~11.1	44~104	300	[31]

## Data Availability

Not applicable.

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
