# Peer review of "Determination of Hexabromocyclododecane in Expanded Polystyrene and Extruded Polystyrene Foam by Gas Chromatography-Mass Spectrometry"

_molecules, 2021, doi:10.3390/molecules26237143_

Round 1

Reviewer 1 Report

I'm sorry, but I'm not happy with the author's responses to my following comments: 3, 4, 9, 10, 13, 15 and 17.

I pointed to the use of the words "determine" or "analyse" instead of "detect" because analytical methods are used to "determine" or "analyse" analyte concentrations, and to use the word "detect" in conjunction with a device (detector) used to "detect" analytes.

I am very dissatisfied with the response to my comment 10: “The study lacks an evaluation of the matrix effect and a results-based answer as to whether it is not necessary to use a matrix-matched calibration for quantification.”, which was:

“Thank you for your suggestion. In this study we optimized the extraction and purification methods and operating conditions (temperature, pressure, and extraction time) for the determination of HBCD in EPS and XPS by GC-MS to avoided matrix effect. We had considered an evaluation of the matrix effect, it is found that it has no effect on the accuracy of the results, this point has been verified by the repeated test results in our laboratory and the test results in different laboratories.”

There is no mention of evaluating the matrix effect in the whole manuscript, but in the lines 84-87 it is stated that: “It is more difficult to detect HBCD in the polystyrene substrates of EPS/XPS than in environmental samples because of matrix effect of EPS/XPS. EPS/XPS is difficult to be removed from the organic solvent remained in the extract dilution and influences the quantification of HBCD.”

Without an exact evaluation of the matrix effect and confirmation that the proposed method is reliable even when using solvent calibration curves without internal standardization, the manuscript is not suitable for publication.

Methods for evaluating the matrix effect can be found in the works:

Ł. Rajski, A. Lozano, A. Uclés, C. Ferrer, A.R. Fernández-Alba, Determination of pesticide residues in high oil vegetal commodities by using various multi-residue methods and clean-ups followed by liquid chromatography tandem mass spectrometry, J. Chromatogr. A, 1304 (2013) 109–120.

Caban, N. Migowska, P. Stepnowski, M. Kwiatkowski, J. Kumirska,  Matrix effects and recovery calculations in analyses of pharmaceuticals based on the determination of β-blockers and β-agonists in environmental samples. Journal of Chromatography A, 1258 (2012) 117–127.

Author Response

Response letter

Note: the comments and responses were listed below in red and black colors, respectively.

Reviewer #1:

I'm sorry, but I'm not happy with the author's responses to my following comments: 3, 4, 9, 10, 13, 15 and 17.

Response Thank you for your comments. We carefully revised the manuscript based on your comments.

Comments (1): Comments on response 3 are as follows: I pointed to the use of the words "determine" or "analyse" instead of "detect" because analytical methods are used to "determine" or "analyse" analyte concentrations, and to use the word "detect" in conjunction with a device (detector) used to "detect" analytes.

ResponseThank you for your comments. We have replaced the misused words in the article according to your suggestion, and you can find these in line 57 and 213.

Comments (2): Comments on response 4 are as follows: Lines 62-63: Instead of “HBCD is generally detected by gas chromatography (GC) [24], liquid chromatography (LC) [25], GC-mass spectrometry (MS) or LC-MS [26].” write “HBCD is generally analyzed by gas (GC) and liquid chromatography (LC) coupled with mass spectrometric (MS) detection […]” Reference [24] is not appropriate as the cited study is on determination of PBDEs and not HBCD.

ResponseThank you for your comments. Reference [24] was misused in the previous article, we have corrected the reference in line 57~60 “Many methods have been developed to determine HBCD concentration in EPS or XPS foam like X-ray fluorescence spectroscopy (XRF), flowing atmospheric pressure afterglow mass spectrometry, liquid chromatography-tandem mass spectrometry (LC-MS/MS) [23-28].”

The quoted content can be found in this work:

Schlummer, M., et al., Rapid identification of polystyrene foam wastes containing hexabromocyclododecane or its alternative polymeric brominated flame retardant by X-ray fluorescence spectroscopy. Waste Management & Research, 2015. 33(7): p. 662-670.

Comments (3): Comments on response 9 are as follows: Lines 91-92, 157-160: Why was the internal standard (IS) not used for calibration? How is it possible to use a solvent calibration curve (without the use of IS) for accurate MS quantification of HBCD in real samples? And what about instrument variability and matrix effect? Calibration has to be performed using IS.

ResponseThank you for your comments.

In this study we aim to develop a convenient,inexpensive and quick method to enable more countries , regions and enterprises to use this method to analyze the HBCD in EPS/XPS. Commonly, the concentration of HBCD in EPS is 0.8%, while the concentration of HBCD in XPS is 4-6%. In such high concentration of HBCD, the external method also can get good analysis results. Comparing to isotope-labeled internal standards, the cost of the external standard could be reduced. The calibration curve was used to quantify the HBCD in real samples and duplicated in five laboratories. Please find these in line 170~176, line 194~198 and line200~203. In addition, external standard method was widely used for the determination of many pollutants. Taking into account that the price of LC-MS/MS is about three times that of gas chromatography mass spectrometers (GC-MS), the popularity of GC-MS may be higher, and more regions and scientific research institutions can use it.

Instrument variability and matrix effect were added in the revised version. You can find this in line 119~126.

Matrix effects for HBCD were determined following the method reported by Caban et al[29]. In order to confirmation that the proposed method is reliable, we randomly select 6 samples of EPS/XPS from different companies and conduct experiments on them. The matrix effect is calculated according to the following formula.

Where A is the peak area of the HBCD standard solution (20mg/L). B is the peak area of the EPS/XPS sample with HBCD standard (20mg/L) added before injection. C is the peak area of the non-spiked EPS/XPS sample. The samples showed acceptable matrix ef-fects ranging from -10.9% to 22.0%.The samples showed acceptable matrix effects ranging from -10.9% to 22.0%. The supplementary experimental data are as follows:

Table 1 Matrix effect experiment results (n=3)

Test material

Matrix effects (%)

EPS1

17.2

EPS2

21.4

EPS3

-10.9

XPS1

13.2

XPS2

22.0

XPS3

5.5

Six laboratories were selected to verify the accuracy of the method, and 10 samples were tested. Experiments are conducted in different laboratories by different experimenters. The interlaboratory relative standard deviation range was 3.68%–9.80%. This proves that there is no problem in quantifying HBCD using the external standard method. The reliability of the method has also been verified.

Figure 1. Results from the six laboratories for EPS (A) and XPS (B).

Comments (4): Comments on response 10 are as follows: Without an exact evaluation of the matrix effect and confirmation that the proposed method is reliable even when using solvent calibration curves without internal standardization, the manuscript is not suitable for publication.

Methods for evaluating the matrix effect can be found in the works:

Ł. Rajski, A. Lozano, A. Uclés, C. Ferrer, A.R. Fernández-Alba, Determination of pesticide residues in high oil vegetal commodities by using various multi-residue methods and clean-ups followed by liquid chromatography tandem mass spectrometry, J. Chromatogr. A, 1304 (2013) 109–120.

Caban, N. Migowska, P. Stepnowski, M. Kwiatkowski, J. Kumirska,  Matrix effects and recovery calculations in analyses of pharmaceuticals based on the determination of β-blockers and β-agonists in environmental samples. Journal of Chromatography A, 1258 (2012) 117–127.

Response

Thank you for your comments. Matrix effect were added in the revised version. The detail results of matrix effect were listed in the response on comment (3).

Comments (5): Comments on response 13 are as follows: Lines 115-116: “For quality control, blank samples were extracted and purified…” Explain closer. Indicate what the blank samples were.

ResponseThank you for your comments. In line 101~102 the blank samples have been specified “An EPS sample without HBCD was selected as the blank sample, extracted and purified together with other EPS/XPS samples.”.

Comments (6): Comments on response 15 are as follows: Lines 170-172: “In the present study, the detection limit of this method is 0.2 mg/kg and the lower limit of detection is 0.8 mg/kg.”    What does the lower limit of detection mean?    In Table 3 the detection limit for the presented method is 0.5 mg/kg and the limit of quantification is 5 mg/kg. So, what is right?

ResponseThank you for your comments. This study followed the Technical Guidance for the Preparation and Revision of the Standard for Environmental Monitoring and Analysis Methods (HJ168-2010), in which the lower limit of the detection average means 4 times the detection limit. But there is no such regulation in the international standard, so in the revised version we deleted the lower limit of detection.

The detection limit obtained by our laboratory was 0.2 mg/kg. In other five laboratories the detection limit was between 0.2 mg/kg and 0.5 mg/kg. So the final detection limit was determined as 0.5 mg/kg on the .

Comments (7): Comments on response 17 are as follows: Lines 199-206: The information can be included in the Introduction section and not here. Here, it would be appropriate to make a comparison with the results of the determination of HBCD in polystyrene from other studies

ResponseThank you for your suggestion. We changed the information to the Introduction section (Line 63~65), and added an estimate of the total amount of HBCD in existing buildings and waste (Line 211~213). The comparison with other studies is placed Line 190~191: “We compared the results from our study with those from previous studies (Table 3). The accuracy of our method met the test requirements.”

Test material

Testing equipment

Concentrations

g/kg

RSDs

%

Limit of quantitation

mg/kg

The literature

EPS/XPS

GC/MS

4.5~33.5

6.64~6.92

0.5

In this study

EPS/XPS

XRF

5~12

7~16

50

[22]

EPS

LC-MS/MS

6.849~7.105

0.2~0.6

0.005

[25]

EPS/XPS

XRF, LC-MS/MS, NMR

6.7~11.1

44~104

300

[31]

Table 2. Comparison of experimental results from different studies.

Reviewer 2 Report

The submitted research is interesting and involves the exploitation of a simple method and more specifically gas chromatography-mass spectrometry method for the determination of hexabromocyclododecane (HBCD) in expanded polystyrene and extruded polystyrene foam. The necessity of this interesting approach is based on the fact that it is crucial to avoid recycling plastic waste with HBCD content exceeding low POP content (LPC) limits. The article is suitable for publication in Molecules.

Overall, the article is well written, the instruments and characterization described is necessary and adequate. The Introduction part is well elaborated and documented and all necessary information and state of the art is given.

A few issues should be encountered prior acceptance.

Part 3.1 should be revisited and more information should be given on how the authors decided to use all these different solvents and their mixtures, how the ratios in the mixtures were decided and solubility parameters should also be given based on the EPS/XPS foams. These data and information are given analytically in the literature. it is very significant for the authors to very well justify all they statements, conclusions and arguments.

The authors mention in Part 3.2: 

"Different injection temperatures will affect the determination of HBCD. When the temperature is too low, it will lead to incomplete gasification of HBCD and reduce the amount available for detection. A temperature that is too high will cause thermal isomerisation and degradation of HBCD diastereomers [29-31]." While they give references for the high temperature they do not mention any literature for the low temperature. Is there any literature? Doe this literature agree with the authors arguments on low temperatures? The authors are encouraged to elaborate further on this issue.

The calibration curve in Figure 2 involves 6 different points. One of them is exceeding too must in order to justify the R2 being 0.9934. What is the marginal error for each measurement?

Part 3.5 is the most important part of the manuscript. It should more analysed and elaborated since it shows that the content of HBCD in EPS/XPS is approximately 1% with a range of 0.45%–0.67% in EPS and 2.98%–3.55% in XPS. The fact that there is a method that can identify the HBCD content is of major importance for impact to society since HBCD in such concentrations is significantly dangerous for human health.

Author Response

Response letter

Note: the comments and responses were listed below in red and black colors, respectively.

Reviewer #2:

The submitted research is interesting and involves the exploitation of a simple method and more specifically gas chromatography-mass spectrometry method for the determination of hexabromocyclododecane (HBCD) in expanded polystyrene and extruded polystyrene foam. The necessity of this interesting approach is based on the fact that it is crucial to avoid recycling plastic waste with HBCD content exceeding low POP content (LPC) limits. The article is suitable for publication in Molecules.

ResponseThank you very much for your support of this work. We carefully revised the manuscript based on your comments. Your comments were listed below in red color and our responses were in black color.

comments (1): Overall, the article is well written, the instruments and characterization described is necessary and adequate. The Introduction part is well elaborated and documented and all necessary information and state of the art is given.

ResponseThank you very much!

comments (2): Part 3.1 should be revisited and more information should be given on how the authors decided to use all these different solvents and their mixtures, how the ratios in the mixtures were decided and solubility parameters should also be given based on the EPS/XPS foams. These data and information are given analytically in the literature. it is very significant for the authors to very well justify all they statements, conclusions and arguments.

ResponseThank you for your comments. Different solvents and their mixtures are selected through literature research. We found that the extraction effect is the best when the mixture ratio is 1:1. In the revised article, we have added relevant study in line 133~137, “Through literature research [23, 26, 30, 31], we chose methanol, toluene, acetone, dichloromethane, n-hexane, n-hexane/isopropanol (1:1, v/v), acetone/methylene chloride (1:1, v/v), toluene/methylene chloride (1:1, v/v), acetone/n-hexane (1:1, v/v) and n-hexane/methylene chloride (1:1, v/v) to conduct experiments on the dissolution of EPS/XPS and the extraction effect of HBCD.”.

Since EPS/XPS cannot be completely dissolved in some solvents, it is difficult for us to give accurate solubility parameters. This study explored the optimal extraction solvent through the method of combining literature and experiment (for the extraction efficiency of HBCD in EPS/XPS foam), which facilitated optimization and sample injection, and reduced the matrix effect.

comments (3): The authors mention in Part 3.2:

"Different injection temperatures will affect the determination of HBCD. When the temperature is too low, it will lead to incomplete gasification of HBCD and reduce the amount available for detection. A temperature that is too high will cause thermal isomerisation and degradation of HBCD diastereomers [29-31]." While they give references for the high temperature they do not mention any literature for the low temperature. Is there any literature? Doe this literature agree with the authors arguments on low temperatures? The authors are encouraged to elaborate further on this issue.

ResponseThank you for your suggestion. We compared the chromatograms of different inlet temperatures, at 190 ℃, the gasification of HBCD was incomplete and the response was low, and at 270 ℃ the HBCD was decomposed.

 Figure 1. Chromatograms of HBCD at different injection temperatures, (a)5mg/L-270℃,(b)5mg/L-230℃,(c)5mg/L-190℃,(d)10mg/L-270℃,(e)10mg/L-230℃,and(f)10mg/L-190℃.

comments (4): The calibration curve in Figure 2 involves 6 different points. One of them is exceeding too must in order to justify the R2 being 0.9934. What is the marginal error for each measurement?

ResponseThank you for your suggestion. We made the calibration curve in Figure 2 according to the standard method. (The Technical Guidance for the Preparation and Revision of the Standard for Environmental Monitoring and Analysis Methods (HJ168-2010)). For the marginal error, the RSD of EPS samples by six laboratories(line 200~203) ranged from 3.78% to 9.76%, and the RSD of XPS samples ranged from 3.68% to 9.80%. The R2 of the calibration curve of all laboratories was between 0.991 and 0.998, which meets the test requirements of the method.

comments (5): Part 3.5 is the most important part of the manuscript. It should more analysed and elaborated since it shows that the content of HBCD in EPS/XPS is approximately 1% with a range of 0.45%–0.67% in EPS and 2.98%–3.55% in XPS. The fact that there is a method that can identify the HBCD content is of major importance for impact to society since HBCD in such concentrations is significantly dangerous for human health.

ResponseThank you for your comments. We have conducted more analysis and elaboration on " we can find the content of HBCD in EPS/XPS is approximately 1% with a range of 0.45%–0.67% in EPS and 2.98%–3.55% in XPS " in line 205~209, “Such high concentration of HBCD may be dangerous to human health [35] and pose a challenge to the treatment of construction waste containing EPS/XPS [27, 28]. Improper handling methods may cause HBCD to leak into the environment, causing serious dam-age to the ecological environment [36].”

Reviewer 3 Report

The manuscript entitled “Determination of Hexabromocyclododecane in Expanded Polystyrene and Extruded Polystyrene Foam by Gas Chromatography-Mass Spectrometry” describes the development and validation of a method to determine the HBCD concentration in different polystyrene samples. While the developed method is satisfactory in terms of analytical parameters such as extraction efficiency and precision, it does not offer significant improvements over other methods. In any case, some questions should be addressed before the manuscript can be considered for publication:

  1. It is recommended to check the English (grammar, use of English, etc.).
  2. Lines 55-56. What is the LPC for HBCD, 100 or 1000 mg/kg? Why is there a range?
  3. Lines 60-62. What are the potential inconveniences of “the most advanced analytical equipment or analytical methods” for determining HBCD? I don´t see how either advanced methods or instruments could hinder the HBCD determination. Indeed, GC-MS is a high-sophisticated instrument. Introduction should be improved in order to highlight all the gaps in the field.
  4. Lines 75-78. Total HBCD concentration can also be quantified by the already reported methods. The determination of HBCD stereoisomers does not hinder the quantification of total HBCD. Indeed, more information is available with them. I suggest to the authors to include a table with information on all the reported methods and critically discuss them in order to argue the advances of the proposed method over the reported ones.
  5. Lines 84-85. Please, give a reference to support this assertion. Environmental matrices are considered “complex matrices”.
  6. Line 109. Standards were prepared in acetone and n-hexane. What is the acetone-n-hexane ratio?
  7. Line 114. The sample amount (0.1g) is small. Have the authors performed any experience to test the representativeness of the whole sample?
  8. Lines 135-137. Procedural blanks should be described in 2.2. sample preparation section
  9. Lines 150-162. The solvent used for polystyrene dissolution is key because it has a potential impact on matrix effects. However, the authors do not offer any data related to the solvent selection.
  10.  Table 3. The authors highlight the fact that the method quantitation limit is almost 10 times lower than previously reported methods. However, samples analyzed have an HBCD concentration range 4.5–6.7 g/kg (EPS) and 29.8–35.5 g/kg (XPS). Thus, the previously reported methods are suitable too for the determination of HBCD in polystyrene foam samples. On the other hand, the authors should indicate the concentration at which analytical parameters (RSD, Recovery, LOQ, R) were calculated in each research article.

As a general suggestion, the authors should give more details about the optimization of sample treatment. In the current format, the manuscript does not show the absence of both thermal degradation and matrix effect as key advantages.

Author Response

Response letter
Reviewer #3: 
The manuscript entitled “Determination of Hexabromocyclododecane in Expanded Polystyrene and Extruded Polystyrene Foam by Gas Chromatography-Mass Spectrometry” describes the development and validation of a method to determine the HBCD concentration in different polystyrene samples. While the developed method is satisfactory in terms of analytical parameters such as extraction efficiency and precision, it does not offer significant improvements over other methods. In any case, some questions should be addressed before the manuscript can be considered for publication.
Response:Thank you very much for your support of this work. We carefully revised the manuscript based on your comments. Your comments were listed below in red color and our responses were in black color. 
comments (1): It is recommended to check the English (grammar, use of English, etc.).
Response:Thank you for your comments. In the revised article, we have checked and corrected the grammar and vocabulary.
comments (2): Lines 55-56. What is the LPC for HBCD, 100 or 1000 mg/kg? Why is there a range?
Response:Thank you for your comments. According to the European Union’s Basel Convention General Technical Guidelines for Persistent Organic Pollutants Waste Management, EU has proposed two concentration limits (100 or 1000 mg/kg) for the low POP content (LPC) limits of HBCD. Until now, the choice of 100 mg/kg or 1000 mg/kg is still under discussion. The Chinese government intend to choose 100 mg/kg as the LPC of HBCD.
comments (3): Lines 60-62. What are the potential inconveniences of “the most advanced analytical equipment or analytical methods” for determining HBCD? I don´t see how either advanced methods or instruments could hinder the HBCD determination. Indeed, GC-MS is a high-sophisticated instrument. Introduction should be improved in order to highlight all the gaps in the field.
Response:Thank you for your comments. Your suggestions are right, and the description was inappropriate. We have revised the articles in line 57~60. As a simple and convenient detection method, this method fully meets the requirements in identifying whether construction waste can be recycled and has a huge application space.
comments (4): Lines 75-78. Total HBCD concentration can also be quantified by the already reported methods. The determination of HBCD stereoisomers does not hinder the quantification of total HBCD. Indeed, more information is available with them. I suggest to the authors to include a table with information on all the reported methods and critically discuss them in order to argue the advances of the proposed method over the reported ones.
Response:Thank you for your comments. Your comments are very valuable. 
The comparison with other studies is placed Line 190~191: “We compared the results from our study with those from previous studies (Table 3). The accuracy of our method met the test requirements.”
Test material Testing equipment Concentrations
g/kg RSDs
% Limit of quantitation
mg/kg The literature
EPS/XPS GC/MS 4.5~33.5 6.64~6.92 0.5 In this study
EPS/XPS XRF 5~12 7~16 50 [22]

EPS LC-MS/MS 6.849~7.105 0.2~0.6 0.005 [25]

EPS/XPS XRF, LC-MS/MS, NMR 6.7~11.1 44~104 300 [31]

Table 3. Comparison of experimental results from different studies.

comments (5): Lines 84-85. Please, give a reference to support this assertion. Environmental matrices are considered “complex matrices”.
Response:Thank you for your comments. Your comments are very valuable. The description of the original sentence is inappropriate, in line 131~132, we revised the article to “Because dissolution of polystyrene from EPS/XPS would cause serious matrix interference in the HBCD analysis, different extraction solvents were screened to optimize the method.”
comments (6): Line 109. Standards were prepared in acetone and n-hexane. What is the acetone-n-hexane ratio?
Response:Thank you for your comments. In line 79~80, we revised the article to “20 mg HBCD was dissolved with 2ml acetone and diluted to 100 ml with n-hexane to prepare the HBCD stock solution (200mg/L).”
comments (7): Line 114. The sample amount (0.1g) is small. Have the authors performed any experience to test the representativeness of the whole sample?
Response:Thank you for your comments. The density of EPS/XPS was only about 20 mg/cm3, and the volume of 0.1 g of samples was about 5 cm3. The EPS/XPS samples were cut into particles smaller than 5 mm, and 0.1 g of samples can represent the whole sample. Please see line 84~85: “The EPS/XPS samples were cut into particles smaller than 5mm. 0.1 g (0.1 mg) of EPS/XPS sample was weighed from the particles above”.
comments (8): Lines 135-137. Procedural blanks should be described in 2.2. sample preparation section
Response:Thank you for your comments. In line 101~102 we revised the article to “An EPS sample without HBCD was selected as the blank sample, extracted and purified together with other EPS/XPS samples.”.
comments (9): Lines 150-162. The solvent used for polystyrene dissolution is key because it has a potential impact on matrix effects. However, the authors do not offer any data related to the solvent selection.
Response:Thank you for your comments. Different solvents and their mixtures are selected through literature research. We found that the extraction effect is the best when the mixture ratio is 1:1. In the revised article, we have added relevant research in line 131~135, “Through literature research [23, 26, 30, 31], we chose methanol, toluene, acetone, dichloromethane, n-hexane, n-hexane/isopropanol (1:1, v/v), acetone/methylene chloride (1:1, v/v), toluene/methylene chloride (1:1, v/v), acetone/n-hexane (1:1, v/v) and n-hexane/methylene chloride (1:1, v/v) to conduct experiments on the dissolution of EPS/XPS and the extraction effect of HBCD.”.
comments (10): Table 3. The authors highlight the fact that the method quantitation limit is almost 10 times lower than previously reported methods. However, samples analyzed have an HBCD concentration range 4.5–6.7 g/kg (EPS) and 29.8–35.5 g/kg (XPS). Thus, the previously reported methods are suitable too for the determination of HBCD in polystyrene foam samples. On the other hand, the authors should indicate the concentration at which analytical parameters (RSD, Recovery, LOQ, R) were calculated in each research article.
Response:Thank you for your comments. In Line 190~191: “We compared the results from our study with those from previous studies (Table 3). The accuracy of our method met the test requirements.”
Table 3. Comparison of experimental results from different studies.
Test material Testing equipment Concentrations
g/kg RSDs
% Limit of quantitation
mg/kg The literature
EPS/XPS GC/MS 4.5~33.5 6.64~6.92 0.5 In this study
EPS/XPS XRF 5~12 7~16 50 [22]

EPS LC-MS/MS 6.849~7.105 0.2~0.6 0.005 [25]

EPS/XPS XRF, LC-MS/MS, NMR 6.7~11.1 44~104 300 [31]

The     Previously reported methods use different detection equipment, such as X-ray fluorescence spectroscopy, nuclear magnetic resonance and LC-MS/MS, each with its own application scenarios. GC/MS provides a new idea for the rapid determination of HBCD in waste, the instrument used is more convenient, and the accuracy and precision also meet the test requirements.

Round 2

Reviewer 1 Report

The authors have made satisfactory changes to the manuscript in response to my comments

Author Response

Note: the comments and responses were listed below in red and black colors, respectively.

Reviewer #1:

The authors have made satisfactory changes to the manuscript in response to my comments.

ResponseThank you very much for your support of this work.

Reviewer 3 Report

 The novelty and scientific impact of this study are still not clearly explained and supported by the results. The previously reported methods met the requirements, too. What is the advantage of the proposed method?

Author Response

Note: the comments and responses were listed below in red and black colors, respectively.

Reviewer #3:

comments (1): The novelty and scientific impact of this study are still not clearly explained and supported by the results. The previously reported methods met the requirements, too. What is the advantage of the proposed method?

ResponseThank you for your comments.

Previously reported methods used different detection equipment, including X-ray fluorescence spectroscopy, nuclear magnetic resonance and LC-MS/MS (Ref 23,26,31). X-ray fluorescence spectroscopy can be used for surface quantification, nuclear magnetic resonance can be used for qualitative analysis.

Comparing the two methods above, LC-MS/MS and GC-MS both can be used for accurate quantitative and qualitative analysis of HBCD. Compared with LC-MS/MS, GC/MS also can achieve accurate quantitative and qualitative analysis of HBCD in EPS/XPS samples, which was studied in our research.

Furthermore, the price of LC-MS/MS is four times that of GC / MS. From this perspective, the popularity of GC/MS is higher, and more laboratories equips GC/MS. Therefore, we developed the GC/MS for the determination of HBCD in EPS/XPS products and wastes.

In addition, although HBCD has recently been banned, EPS/XPS products containing HBCD are still used on the surface of buildings and will last for 20-50 years until the demolition of buildings. In the future, HBCD in EPS/XPS wastes generated from the demolition of these buildings needs to be detected and analyzed. Therefore, the detection of HBCD in EPS/XPS samples by GC/MS still has great application value.

You can find this in line 224~228.
